# Major Histocompatibility Complex (MHC) Diversity of the Reintroduction Populations of Endangered Przewalski’s Horse

**DOI:** 10.3390/genes13050928

**Published:** 2022-05-23

**Authors:** Yongqing Tang, Gang Liu, Shasha Zhao, Kai Li, Dong Zhang, Shuqiang Liu, Defu Hu

**Affiliations:** 1College of Ecology and Nature Conservation, Beijing Forestry University, Beijing 100085, China; tangyq06@126.com (Y.T.); bethy1210@163.com (S.Z.); likai_sino@sina.com (K.L.); ernest8445@163.com (D.Z.); shuqiangliu@163.com (S.L.); 2Institute of Wetland Research, Chinese Academy of Forestry, Beijing Key Laboratory of Wetland Services and Restoration, Beijing 100091, China

**Keywords:** Przewalski’s horse, major histocompatibility complex (MHC), reintroduction, genetic diversity, single strand conformation polymorphism (SSCP), DQA genes

## Abstract

Major histocompatibility complex (MHC) genes are the most polymorphic in vertebrates and the high variability in many MHC genes is thought to play a crucial role in pathogen recognition. The MHC class II locus DQA polymorphism was analyzed in the endangered Przewalski’s horse, *Equus przewalskii*, a species that has been extinct in the wild and all the current living individuals descend from 12 founders. We used the polymerase chain reaction-single strand conformation polymorphism (PCR-SSCP) to detect the polymorphism within the MHC DQA in 31 Przewalski’s horses from two reintroduced populations. Consequently, only seven alleles were identified, with only four presenting in each population. In comparison with other mammals, the Przewalski’s horse demonstrated less MHC variation. The nucleotide genetic distance of the seven ELA-DQA alleles was between 0.012 and 0.161. The Poisson corrected amino acid genetic distance of the founded alleles was 0.01–0.334. The allele and genotype frequencies of both reintroduced populations of Przewalski’s horse deviated from the Hardy–Weinberg equilibrium. Specific MHC DQA alleles may have been lost during the extreme bottleneck event that this species underwent throughout history. We suggest the necessity to detect the genetic background of individuals prior to performing the reintroduction project.

## 1. Introduction

The major histocompatibility complex (MHC) is a hypergene family with the highest polymorphism in vertebrates [1,2]. Its expression product is the core part of the immune system, which presents the invading pathogen peptides to the surface membrane of T cells and initiates immune response [3]. The formation and maintenance of MHC gene polymorphism are the result of heterozygote advantage and pathogen-mediated selection [4]. Heterozygous MHC genes can bind and present more kinds of antigenic peptides, and have stronger abilities to resist pathogen invasion compared with homozygous ones [5]. Thus, the heterozygous individuals are preferred by natural selection [6]. MHC heterozygote individuals are able to recognize more parasite-derived peptides. Sexual selection, alongside natural selection, plays an important role in maintaining MHC polymorphism [4,7].

The host–parasite coevolution leads to an increase in the frequency of rare alleles in the population, and this frequency selectively drives the high level of MHC polymorphism [8,9]. In addition, animals prefer individuals whose MHC gene is different from themselves in mate selection, which is conducive to the increasing heterozygosity of the offspring’s MHC gene [3,10]. Moreover, the survival of the fetus can be increased by carrying antigens that are different from the mother, which may contribute to the formation of MHC gene polymorphism [11].

The MHC gene has been considered as an important functional molecular marker, which has unique advantages and application prospects in conservation genetics [8,9]. The MHC gene is characterized by high variability, which is related to population viability, population fecundity and disease and pathogen resistance [12,13,14]. The MHC of mammals includes a gene family that is typically divided into the following three main types: class I, II, and III. Class II genes are expressed mainly on immune cells and function in detecting bacterial and parasitic antigens from the extracellular environment [4,15,16].

Przewalski’s horse *Equus przewalskii* is the only real existing wild horse in the world, which is a representative species in the arid environment [17]. With an evolutionary history of 60 million years, Przewalski’s horse has the characteristics of strong running ability, which is known as the “living fossil” of important biological significance. Przewalski’s horse is listed among the wildlife under first-class national protection in China. The original habitat of Przewalski’s horse was gradually destroyed by human activities, and due to habitat loss and hunting, its wild population went extinct in China and Mongolia in the middle of last century. The current populations are the offsprings of 12 captive Przewalski’s horses (Boyd, 1994). In previous studies, several researchers have used different molecular markers to assess the genetic diversity and genetic structure of reintroduced Przewalski’s horses, including mitochondrial DNA [18,19], microsatellites [20,21], and the whole genomics [22,23]. The MHC diversity is investigated in populations of Przewalski’s horses in Europe [24]; however, it is still not known about the MHC diversity for the reintroduction populations in China.

In this study, the genetic diversity of MHC genes of two reintroduced populations of Przewalski’s horses were studied. The second exon of the DQA site was selected because this site is located in the α Chain, encoding MHC-II molecule α1 functional proteins. By combining blood, hair and fecal samples, we used the polymerase chain reaction-single strand conformation polymorphism (PCR-SSCP) to detect the polymorphism within the MHC DQA in 31 Przewalski’s horses. The work would allow us to assess the genetic diversity of Przewalski’s horses at the MHC DQA locus, and compare the allele frequency difference between two reintroduced populations.

## 2. Materials and Methods

### 2.1. Sample Collection

A total of 31 samples of Przewalski’s horses were obtained from two reintroduced populations (Table 1). Eleven samples were collected from Przewalski’s Horse Breeding Research Center (PHB) in Xinjiang Province, China, including two hair samples and nine blood samples. Twenty fecal samples were gathered from Endangered Animal Breeding Research Center (EAB) in Gansu Province, China. PHB was established in 1985, and is located 40 km northwest of jimusar County, Xinjiang, and its geographical coordinates are E 88°44′26″ and N 44°12′12″, with an average altitude of 580 m. The current population size in PHB is 274, which originated from the 16 reintroduced individuals from Germany and Britain through the reintroduction project initiated in 1986 [25]. EAB is located in the southern edge of Tengger Desert, Wuwei City, Gansu Province, which was established in 1987. The geographical coordinates of EAB are E 102°52′50″ and N 37°52′52″, with an average altitude of 200 m. EAB is one of the most important national research centers, which is responsible for the reintroduction, breeding and wild release of desert endangered wild animals, which now hosts 84 individuals of Przewalski’s horse.

In the process of fecal sample collection, we firstly observed the behavior of Przewalski’s horses, and identified each individual by the label on its body. Fresh fecal samples were collected and stored in sterile plastic tubes that were sterilized by UV. By adding 100% ethanol, the fecal samples were preserved in the lab before DNA extraction. The hair and blood samples were preserved at −20 °C.

### 2.2. DNA Extraction and PCR

Blood and hair genomic DNA were isolated and purified by proteinase K digestion and phenol/chloroform extraction following standard procedures [26]. Genomic DNA in the fecal samples were extracted based on the method published in the literature [18].

The reverse oligonucleotide primer (5′-CTGATCACTTGCCTCCTATG-3′) and the forward primer (5′-TGGTAGCAGCAGTAGTGTG-3′) were designed from the consensus sequences of DQA among different mammals. The polymerase chain reaction (PCR) was performed with a total of 25 μL containing 2.5 μL of 10 × buffer (without Mg^2+^), 0.8 μM of dNTP, 1.5 μM of MgCl_2_, 0.5 μM of each primer, 0.5 units of Taq DNA polymerase (Takara), 0.5 units of bovine serum protein (BSA, Takara, Shanghai, China); 80 ng of template DNA. The amplification was conducted using the steps of 5 min at 96 °C followed by 31 cycles of 1 min at 96 °C, 1 min at 55 °C, 1 min at 72 °C, and then followed by 10 min at 72 °C. All the PCR products were detected using a gel agarose, and the products of good quality were stored at 4 °C.

### 2.3. SSCP Analysis and Cloning

To perform denaturing single-stranded conformation polymorphism (SSCP) electrophoresis [27], a mixture of 6 μL of the purified PCR products, 4 μL of the denaturing loading dye (95% deionized formamide, 20 mM Na_2_ EDTA, 0.05% xylene cyanol and 0.05% bromo-phenol blue), and 10 μL of H_2_O, were heated at 95 °C for 10 min, and then cooled in ice water for 15 min prior to being loaded onto SSCP gels. The SSCP gels for DQA typing consisted of 30% *v*/*v* Acr-Bis (37.5:1), 0.5 × TBE (44.5 mm Tris, 44.5 mm boric acid, 1 mm Na_2_ EDTA, pH = 8.4), 0.05% *v*/*v* TEMED; 0.05% *w*/*v* ammonium persulfate (AP). Electrophoresis was carried out in 0.5 × TBE for 14 h at 10 °C by 600 V. The SSCP patterns were visualized by staining gels for 15 min in 0.5 × TBE with 0.7 μL/mL of ethidium bromide. The alleles of the homozygous individuals could be directly detected on 1.5% agarose gel, then tapped and recycled with a DNA gel Recovery Kit (Takara). The alleles of heterozygote individuals were first separated by polyacrylamide gel, then recovered by polyacrylamide gel, and 4 μL of the recovered product was taken for PCR. The recycled DNA of different alleles were connected with a PGM-T vector, and transformed into *E. coli* TOP10 strain. The positive clones were screened by PCR with the original amplified primers (the PCR reaction was the same as above). The positive clones were analyzed by SSCP with the genomic amplification products as the control, and the positive clones with an electrophoretic migration distance equal to the genomic control were sent for sequencing. A sequence was accepted if at least three reactions produced identical results at a given base. The sequences were aligned and edited using the Clustal x program [28].

### 2.4. Statistical Analysis

Different alleles of DQA of Przewalski’s horse were translated into protein sequences by MEGA [29], and the average nucleotide genetic distance (Kimura 2-parameter model) and amino acid genetic distance (Poisson correction) were calculated. The standard error was obtained by 1000 repeated tests. By comparing with the protein sequences of human homologous genes, the antigen binding site (ABS) and non-antigen binding site (non-ABS) of the protein sequence encoded by the DQA gene of the Przewalski’s horses were inferred. The allele frequencies of the DQA loci of different populations were determined by Genepop software [30]. Observed heterozygosity (*H**o*), expected heterozygosity (*He*) and Hardy–Weinberg equilibrium were calculated. By comparing the sequenced different alleles of DQA of Przewalski’s horses with other equine sequences recorded in GenBank based on the Kimura 2-parameter adjacency method (bootstrap sampling 1000 times) with MEGA, the phylogenetic tree of DQA was constructed, including human and bovine gene sequences as the outgroup.

## 3. Results

In 31 samples of Przewalski’s horse, a total of 7 PCR-SSCP defined MHC DQA alleles were detected. Among those alleles, ELA-DQA3, ELA-DQA4 and ELA-DQA1 were the same as the sequences of ELA-DQA*0501, ELA-DQA*0601 and ELA-DQA*0701 published in GenBank, respectively, which means that the other four alleles were newly identified in Przewalski’s horses in this study (Figure 1). The amino acid sequences that omit the primer sites are given in Figure 1, along with the corresponding nucleotide sequences for variable positions. The amino acid positions, documented using X-ray crystallography to be important in the antigen binding site (ABS) [31], were indicated by a solid point in Figure 1. The amino acid differences among the MHC alleles of Przewalski’s horses ranged from 1 to 15 amino acid residues, with an average of 7.0, accounting for 8.75% of the total sequence length.

We found four alleles in PHB, and four alleles in EAB. The allele frequency was estimated for each population, and the results showed that ELA-DQA1 was the dominant allele in EAB with a high gene frequency of 0.65, while it was absent in PHB. For PHB, the allele ELA-DQA3 and ELA-DQA4 were dominant alleles, with the gene frequency being 0.3636 (Table 2). The fitness chi-square test showed that the two populations of wild horses did not reach Hardy–Weinberg equilibrium at this locus (χ^2^ < 0.05).

Homozygosity (*Ho*), heterozygosity (*He*), polymorphism information contents and the number of effective alleles were estimated. The results indicated that the *He* value of the PHB population (0.69) was higher than those of the EAB population (0.54). For *PIC*, the PHB population was also higher than that in the EAB population, and the same for *Ne* (Table 3). All of those metrics suggested that the PHB population had more MHC polymorphism than the Gansu EAB population.

The Kimura 2-parameter nucleotide genetic distance between the seven alleles ranged from 0.012 to 0.161. The genetic distance of amino acids ranged from 0.012 to 0.334 (corrected by Poisson). The variation in the amino acid residues ranged from 1 (1.25%) to 15 (18.75%). The genetic distance between the gene antigen binding sites and non-antigen binding sites is shown in Table 4.

The neighbor-joining tree was constructed using our identified DQA allele sequences of Przewalskl’s horses, by combing nine other equine sequences published in the literature. It could be observed from the phylogenetics tree that ELA-DQA2 and ELA-DQA3 in Przewalski’s horses clustered together, but separated from ELA-DQA5 and ELA-DQA6. ELA-DQA1 and ELA-DQA7 belonged to another branch (Figure 2). The phylogenetic relationship between zebras and domestic horses was not as close as that between Przewalski’s horses and domestic horses. Domestic horses were closer to the tree roots than other equine animals.

## 4. Discussion

Compared with other mammals in the observed heterozygosity, such as the giant panda (*Ailuropoda melanoleuca*, *Ho* = 0.48∼0.73) [32], the golden snub-nosed monkey (*Rhinopithecus roxellana*, *Ho* = 0.53) [33], the Przewalski’s horse (*Ho* = 0.39) showed very low genetic diversity at the MHC locus of DQA. The Przewalski’s horse experienced several serious bottleneck events in the mid-20th century [34]. After more than 100 years of human reproduction and breeding, the wild horse was reintroduced into its original habitats. The history of bottlenecks and declining population size are the main reasons behind the low diversity and high inbreeding [35,36]. The genetic diversity level is still impacted by the founder effect, which means that the source population has already presented low levels of genetic variation [37]. The amino acid difference between the MHC alleles of Przewalski’s horses was 17.2, which is much less than that inKonik Polski horses(27) [38], and domestic horses [38,39,40]. The low MHC genetic diversity in Przewalski’s horses is also supported by previous studies based on different molecular markers. For example, low expected heterozygosity was also observed based on microsatellites, and a high level of inbreeding coefficient of 0.19 was reported [20].

The genetic diversity of the DQA locus of the PHB population was higher than that in the EBA population. This may be the result of the planned grouping and isolation of wild horses by Xinjiang Przewalski’s horse breeding center for many years to avoid inbreeding [20]. The Hardy–Weinberg equilibrium test shows that both the Xinjiang population and the Gansu population do not meet the Hardy–Weinberg equilibrium, which may be related to the fact that the two captive populations of Przewalski’s horses in China have been reestablished by a small number of individuals with some alleles in the parental population. It is caused by the lack of mating and reproduction with other Przewalski’s horse groups and the small genetic difference between them. The gene loss and random fluctuation of gene frequency in the small population of Przewalski’s horses were also the reasons for the Hardy–Weinberg imbalance of the Przewalski’s horse population.

The consequence of low MHC diversity in Przewalski’s horses is the observed heavy load of parasites [41]. Parasites can become a direct factor, leading to population collapse, which poses a serious threat when reintroducing small populations [42]. For example, parasite infection of Przewalski’s horses is 3–5 times higher than that of Mongolian wild donkeys and domestic horses in the same region [43]. It can be inferred that low MHC diversity may suggest that the genetic basis does not function well in resisting those parasites. The genetic distance between the MHC alleles is particularly important for a population with a limited number of alleles, because it helps to improve the genetic diversity in the population, which is crucial to increase the resistance to different pathogens. Therefore, when facing a variety of exogenous pathogens, the disease susceptibility of Przewalski’s horses may be significantly higher than that of other vertebrates.

In the process of the reintroduction project, it is necessary to detect the genetic background of individuals [44]. On one hand, individuals carrying rare alleles can be identified and given more care [45]. On the other hand, inbreeding reproduction can be avoided or reduced to some extent via both pedigree records and genetics approach. For example, the allele frequency difference between the two populations indicates that the breeding management strategy has changed the distribution frequency of the MHC gene of Przewalski’s horses, making some alleles present in one population, but other alleles absent in another population. As the diversity of MHC genes is directly related to the immune function, the loss or absence of some particular alleles may reduce the resistance to some diseases. Therefore, in view of the sustainable development of the Przewalski’s horse population, it is necessary to improve the reproductive rate of Przewalski’s horses carrying rare alleles.

Future conservation concerns regarding Przewalski’s horses should depend on the genetics method, followed by scientific genetic diversity guidelines and non-invasive monitoring in the population development and management [46]. Overall, a preliminary study on the MHC polymorphism of Przewalski’s horses was conducted, so as to provide a certain theoretical basis for the scientific management of Przewalski’s horses. However, there are still some problems to be solved focusing on MHC genes, such as the relationship between the immune level and MHC polymorphism, the relationship between parasite infection and MHC diversity. In addition, the specific reintroduction projects and breeding managements would be influenced by some practical factors, for example, the gene flow between two populations (EAB and PHB) is quite limited, due to the difficulty in genetic exchange, because those two populations are distributed in two different sites located in different provinces, and they are managed by different centers. So, we strongly recommend interbreeding between different facilities, which should be set as the action priority to reduce inbreeding and enhance adaptive potential.

## Figures and Tables

**Figure 1 genes-13-00928-f001:**
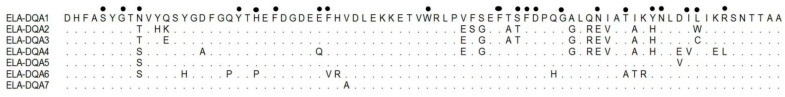
Amino acid sequences of MHC DQA alleles detected in the Przewalski’s horses. The amino acid residues were given in the international single letter code. A dot indicated that the amino acid was the same as in the consensus sequence. A solid point denoted an antigen binding site.

**Figure 2 genes-13-00928-f002:**
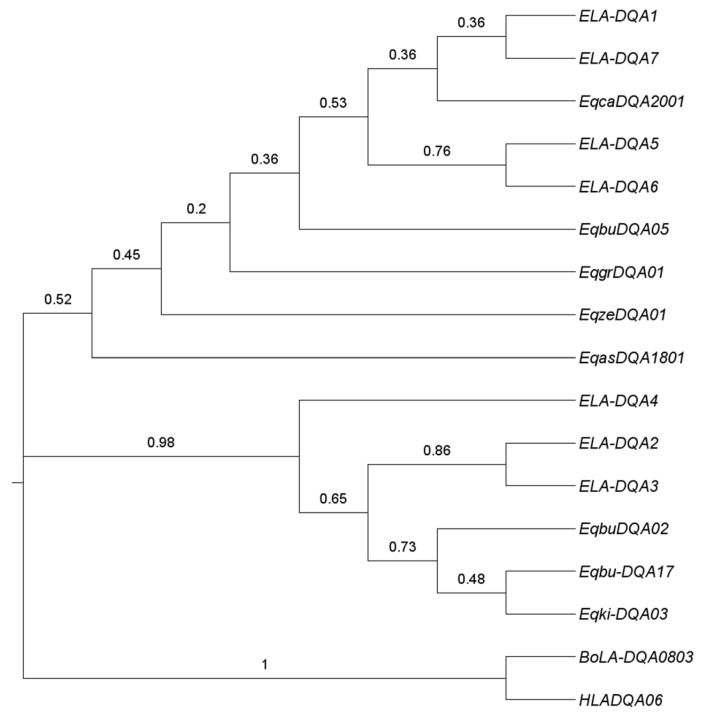
Phylogenetics tree among different equine species based on MHC ELA-DQA exon2 alleles. The human and bovine gene sequences were set as the outgroup. Eqca: *Equus caballus*; Eqbu: *Equus burchellii*; Eqgr: *Equus grevyi*; Eqze: *Equus zebra*; Eqas: *Equus africanus*; Eqki: *Equus kiang*; BoLA: *Bos taurus*; HLA: *Homo sapiens*. The number above the branch indicates the bootstrap value.

**Table 1 genes-13-00928-t001:** Summary information on the samples of Przewalski’s horse.

Population	Sample Type	Sample Size	Sampling Year
PHB	Hair	2	2009
	Blood	9	2009
EAB	Faces	20	2015

**Table 2 genes-13-00928-t002:** The allelle frequencies for two populations of Przewalski’s horse.

Population	ELA- DQA1	ELA- DQA2	ELA- DQA3	ELA- DQA4	ELA- DQA5	ELA- DQA6	ELA- DQA7	*p* Value
PHB	0.00	0.23	0.36	0.36	0.00	0.05	0.00	0.0033
EAB	0.65	0.00	0.20	0.00	0.13	0.00	0.03	0.00

**Table 3 genes-13-00928-t003:** Homozygosity, heterozygosity, polymorphism information contents and the number of effective alleles of ELA-DQAexon2 in two populations of Przewalski’s horses.

Population	*Ho*	*He*	*PIC*	*Ne*
PHB	0.32	0.69	0.62	3.14
EAB	0.47	0.54	0.42	1.86
Average	0.39	0.61	0.52	2.5

**Table 4 genes-13-00928-t004:** The average nucleotide and amino acid distances among the Przewalski’s horse MHC DQA alleles.

Kimura 2-Parameter			Amino Acid Genetic Distance (Poisson Correction)		
All Sites	ABS	Non-ABS	All Sites	ABS	Non-ABS
8.5	19.7	5.5	17.2	38.7	11.7

## Data Availability

Not applicable.

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
