# Peer review of "Major Histocompatibility Complex (MHC) Diversity of the Reintroduction Populations of Endangered Przewalski’s Horse"

_genes, 2022, doi:10.3390/genes13050928_

Round 1

Reviewer 1 Report

The authors describe genotyping of major histocompatibility complex by PCR-SSCP and sequencing in Przewalski’s horse bred in China.

Several alleles were reported, and their frequencies and other characteristics were calculated. 

Outlook to further application of the results is given.

The suggested minor modifications are below.

Line 13

’…living individuals descend from 12 founders a species.’

delete ’a species’

Line 44

’…animals prefer individuals whose MHC gene is different from themselves in mate selection…’

Here the authors refer to Huang and Pemberton 2021 as well. Their article title is about providing little support for post-copulatory selection on major histocompatibility complex haplotypes. The text should reflect that ’little support’.

line 62

’…first-class national protection.’

In China I presume. Please add the country to this sentence.

Section DNA extraction and PCR:

Mg2+:  2+ should appear as uppercase.

Cl2: 2 should appear as lowercase.

section 2.3:

H2O, Na2EDTA: 2 should appear as lowercase.

line 132

’…recovered by polyacrylamide gel…’

recovered from

line 137

’… with matched bands were…’

Please change to

…with electrophoretic migration distance equal to the genomic control were…

line 149

Ho…He

line 160

’…omitting the primers were…’

omitting the primer sites were

Figure 2.

Please resolve acronyms like BoLa, HLA, Eqas, Equ,… in the subscription of the figure, and give the meaning of the numbers on the branches as well.

Line 205

’ Compared with other mammals…’

Give numbers obtained from panda and monkey.

line 215

’… is much less…’

How much is that?

line 219

’… was reported…’

Reported numbers are missing.

Line 225

Hardy-

line 229

’… Although the number of this population will increase later, however, it is caused…’

Please delete the part

’… Although the number of this population will increase later, however,…’

Author Response

#Reviewer 1

Q1: Line 13’…living individuals descend from 12 founders a species.’

delete ’a species’

R1:“a species”  has been deleted in the revised manuscript.

Q2: Line 44 ’…animals prefer individuals whose MHC gene is different from themselves in mate selection…’Here the authors refer to Huang and Pemberton 2021 as well. Their article title is about providing little support for post-copulatory selection on major histocompatibility complex haplotypes. The text should reflect that ’little support’.

R2: Thank you very much for this valuable comment. The Huang and Pemberton 2021 reference has been replaced with another reference,Burger et al. 2017, which tested the hypothesis that female horses were more likely to become pregnant if exposed to an MHC-dissimilar than to an MHC-similar male. So the new reference strongly supports our statement here.

Q3:line 62’…first-class national protection.’In China I presume. Please add the country to this sentence.

R3:Yes here it’s in China. So we have added it in this sentence. 

Q4: Mg2+:  2+ should appear as uppercase. Cl2: 2 should appear as lowercase.

R4: We have revised them and double checked other similar issues in the manuscript.

Q5: H2O, Na2EDTA: 2 should appear as lowercase.

R5: We have revised them and double checked other similar issues in the manuscript.

Q6: line 132’…recovered by polyacrylamide gel…’

recovered from

R6: “recovered by”has been replaced with “recovered by”.

Q7:line 137’… with matched bands were…’Please change to

…with electrophoretic migration distance equal to the genomic control were…

R7:Thanks so much, we have revised it based on your comment.

Q8:line 149 Ho…He

R8:Those two symbols have been changed.

Q9:line 160’…omitting the primers were…’omitting the primer sites were

R9: We have revised it based on your comment.

Q10:Figure 2.     Please resolve acronyms like BoLa, HLA, Eqas, Equ,… in the subscription of the figure, and give the meaning of the numbers on the branches as well.

R10: We have provided detailed information about those acronyms.Eqca: Equus caballus; Eqbu: Equus burchellii; Eqgr: Equus grevyi; Eqze: Equus zebra; Eqas: Equus africanus; Eqki: Equus kiang; BoLA: Bos taurus; HLA: Homo sapiens. The number above the branch indicates the bootstrap value.

Q11:Line 205’ Compared with other mammals…’Give numbers obtained from panda and monkey.

R11:We have rephrased this sentence by saying like below:

Compared with other mammals in observed heterozygosity, such as the giant panda (Ailuropoda melanoleuca, Ho = 0.49-0.73) (Wan et al. 2006), the golden snub-nosed monkey (Rhinopithecus roxellana, Ho = 0.53) (Zhang et al. 2018), Przewalski’s horse (Ho = 0.39) showed very low genetic diversity at the MHC locus of DQA. 

Q12:line 215’… is much less…’How much is that?

R12: The amino acid difference between MHC alleles of Przewalski’s horse was 7. So we have revised this sentence. The amino acid difference between MHC alleles of Przewalski’s horse was 17.2, which is much less than that in Konik Polski horses (27) (Jaworska et al. 2020), and domestic horse (35) (Viļuma et al. 2017).

Q13: line 219’… was reported…’Reported numbers are missing.

R13: In the reference (Liu et al. 2014), the inbreeding coefficient of Przewalski’s horse was 0.19, so we provided and cited the number here.

Q14: Line 225Hardy-line 229 ’… Although the number of this population will increase later, however, it is caused…’Please delete the part

’… Although the number of this population will increase later, however,…’

R14: We have deleted it based on your comments.

Reviewer 2 Report

The authors present an analysis of the genetic diversity present in the ELA-DQA gene in the endangered Przewalski's horse, and show that these genetic variants are not in Hardy Weinburg equilibrium, suggesting that strong bottlenecks in the population history may have reduced the diversity present in the current population.

I believe that the reason for using SSCP to identify variants needs to be better explained by the author: why not just use PCR to amplify the region of interest, then sequence this? My guess would be that the authors want to identify haplotypes, but this could also be done with sufficiently long sequence reads (the region shown in Figure 1 is 80 amino acids, so 240 base pairs, long, which would easily be covered by 2×150bp paired-end sequence).

How closely related are the two Chinese populations? I realise that since all current animals are the descendents of only 12 ancestors, they must be quite closely related, but is there any recent interbreeding between the two? What about between these two and the European or other populations around the world? If there is any population mixture, what impact would this have on the interpretation of the results presented here?

Both figures are blurry enough to be difficult to read; I would recommend using higher resolution images so they can be shown at a larger size without losing clarity (although this could just be an artifact from making the draft PDF file).

Figure 1 would be improved by including the other sequences that are included in the tree in Figure 2.

Figure 2 (the phylogenetic tree) would benefit from labels that more clearly identify which alleles come from which species.

In several places in the manuscript, the name "Przewalski's" has been misspelt as "Przewalskll’s". Please correct these.

Author Response

#Reviewer 2

Q1: I believe that the reason for using SSCP to identify variants needs to be better explained by the author: why not just use PCR to amplify the region of interest, then sequence this? My guess would be that the authors want to identify haplotypes, but this could also be done with sufficiently long sequence reads (the region shown in Figure 1 is 80 amino acids, so 240 base pairs, long, which would easily be covered by 2×150bp paired-end sequence).

R1: Thank you very much for your important comments. We totally agree with you that haplotypes can be identified by using the next-generation sequencing, since the 2×150 bp paired-end sequencing can cover it. For example, there are some references using this method. In our manuscript, we used the traditional method, SSCP, to identify MHC variants when we were preparing the experiment, and a lot of references have used SSCP in MHC related studies. Besides SSCP,we used the clone sequencing to double identify the MHC alleles in our study. So combing SSCP and clone sequencing could work successfully to genotype the MHC loci, and most importantly this approach has the advantage of low cost.

Q2: How closely related are the two Chinese populations? I realize that since all current animals are the descendents of only 12 ancestors, they must be quite closely related, but is there any recent interbreeding between the two? What about between these two and the European or other populations around the world? If there is any population mixture, what impact would this have on the interpretation of the results presented here?

R2: Your question is very good. Between 1985 and 1994, 22 Przewalski’s horses from German and American zoos were imported to EAB, while another 18 Przewalski’s horses from western zoos (Germany, England, and United States) were sent to PHB. So the reintroduction projects made it a little bit different for the founders between EAB and PHB, but actually all living individuals were derived from 12 ancestors, so the genetic distance between those two populations is not high, which could be seen from our previous study (Liu et al. 2014) based on microsatellites. Unfortunately, the interbreeding between those two populations (EAB and PHB)  is quite limited due to the difficulty in practice, because those two populations are located in two different sites located in different provinces in China, and they are managed by different centers. Focusing on this point, we added two sentences talking about the importance of genetic exchange between different breeding centers in the reintroduction success of Przewalski’s horse. Please see the last paragraph of the revised manuscript.

Q3: Both figures are blurry enough to be difficult to read; I would recommend using higher resolution images so they can be shown at a larger size without losing clarity (although this could just be an artifact from making the draft PDF file).

R3: We have made the figure 1 again, and hopefully its resolution will be suitable for the publication. At least it is very clear when being put in the word. It is possible sometimes that the resolution will be degraded by transferring to the format PDF by the submission system. We can provide the original figure in the format Tiff, if needed.

Q4: Figure 1 would be improved by including the other sequences that are included in the tree in Figure 2.

R4: Thank you so much for your suggestion. We have included the sequences from other species into Figure 1.

Q5: Figure 2 (the phylogenetic tree) would benefit from labels that more clearly identify which alleles come from which species.

R5: We have provided detailed information about those acronyms.Eqca: Equus caballus; Eqbu: Equus burchellii; Eqgr: Equus grevyi; Eqze: Equus zebra; Eqas: Equus africanus; Eqki: Equus kiang; BoLA: Bos taurus; HLA: Homo sapiens. The number above the branch indicates the bootstrap value.

Q6: In several places in the manuscript, the name "Przewalski's" has been misspelt as "Przewalskll’s". Please correct these.

R6: We have corrected it, and double checked others in the revised manuscript.
